# Incidence and predictors of mortality within the first year of antiretroviral therapy initiation at Debre-Markos Referral Hospital, Northwest Ethiopia: A retrospective follow up study

Agazhe Aemro[ID][1]*, Mulugeta Wassie[1], Basazinew Chekol[ID][2]

1 Department of Medical Nursing, School of Nursing, College of Medicine and Health Science, University of Gondar, Gondar, Ethiopia, 2 Department of Anesthesiology, College of Medicine and Health Science, Debre-Tabor University, Debre-Tabor, Ethiopia

* agazhe1049@gmail.com

**Data Availability Statement:** All relevant data are within the manuscript and its Supporting information files. Similarly, the minimally

## Abstract

### Background

Acquired Immunodeficiency Syndrome (AIDS) is one of the most fatal infectious diseases in the world, especially in Sub-Saharan Africa, including Ethiopia. Even though Antiretroviral therapy (ART) significantly decreases mortality overall, death rates are still highest especially in the first year of ART initiation.

### Objective

To assess the incidence and predictors of mortality within the first year of ART initiation among adults on ART at Debre-Markos Referral Hospital, Northwest Ethiopia.

### Methods

A retrospective follow-up study was conducted among 514 newly enrolled adults to ART from 2014 to 2018 at Debre-Markos Referral Hospital. Patients' chart number was selected from the computer using a simple random sampling technique. Data were entered into EPI-INFO 7.2.2.6 and analyzed using Stata 14.0. The mortality rate within the first year was computed and described using frequency tables. Both bivariable and multivariable Cox-proportional hazard models were fitted to show predictors of early mortality.

### Results

Out of 494 patient records included in the analysis, a total of 54 deaths were recorded within one year follow-up period. The overall mortality rate within 398.37 person years (PY) was 13.56 deaths/100 PY with the higher rate observed within the first three months. After adjustment, rural residence (Adjusted Hazard Ratio (AHR) = 1.97; 95% CI: 1.05–3.71), ≥ 6 months pre-ART duration (AHR = 2.17; 95% CI: 1.24–3.79), ambulatory or bedridden

Anonymized dataset has been uploaded as a Supporting information file.

**Funding:** The author(s) received no specific funding for this work.

**Competing interests:** The authors have declared that no competing interests exist.

**Abbreviations:** 3TC, Lamivudine; AHR, Adjusted Hazard Ratio; AIDS, Acquired Immunodeficiency Syndrome; ART, Antiretroviral Therapy; BMI, Body Mass Index; CD4, Cluster of Differentiation Four; CHR, Crude Hazard Ratio; CPT, Cotrimoxazole Prophylactic Therapy; EFV, Efavirinz; HAART, Highly Active Antiretroviral Therapy; HIV, Human Immunodeficiency Virus; IPT, Isoniazid Preventive Therapy; IQR, Inter Quartile Range; IRIS, Immune Reconstitution Inflammatory Syndrome; OIs, Opportunistic Infections; PY, Peron Year; TDF, Tenofovir Disoproxil Fumarate; WHO, World Health Organization.

functional status at enrolment (AHR = 2.18; 95% CI: 1.01–4.74), and didn't take Cotrimoxazole preventive therapy (CPT) during follow-up (AHR = 1.88; 95% CI: 1.04–3.41) were associated with early mortality of adults on ART.

## Conclusion

Mortality within the first year of ART initiation was high and rural residence, longer pre-Art duration, ambulatory or bedridden functional status and didn't take CPT during follow-up were found to be independent predictors. Hence, giving special attention for patients from rural area and provision of CPT is crucial to reduce mortality.

## Introduction

An infection with Human Immunodeficiency Virus (HIV) can lead to AIDS which results in a gradual decline and failure of the immune system. Once the immune system became compromised, the patient becomes highly susceptible to life-threatening infection and results in early death. HIV/AIDS is one of the most fatal infectious diseases in the world, especially in Sub-Saharan Africa, it had a massive impact on health outcomes and life expectancy [1].

Even though there is a substantial decline in mortality of patients with AIDS, still many people with the disease were dying. Based on Global HIV/AIDS statistics of 2020 fact sheets, in 2019, around 690,000 people died from AIDS-related illnesses worldwide [2]. Now a day treatment for patients with HIV infection is available. But HIV-infected individuals, especially in Africa including Ethiopia, still hesitates to start the treatment early which results in progression of the disease to the advanced stage. This intern results in risk of early mortality for patients with HIV [3–5].

In Ethiopia, HIV-infected patients hesitate to start ART as early as possible due to different reasons. Of these, fear of stigma in the broader community is the most common reason to delay in receiving treatment. Additionally, lack of knowledge about the benefits of ART and belief that using ART was prohibited by God were other factors contributing to hesitate in using ART. But, it could be overcome with continuous education and counseling of patients about the benefits of ART and U = U (undetectable = un-transmittable). Education of patients about U = U has numerous advantages and maximizes the wellbeing of people living with HIV. It offers attainment of viral suppression, strengthening patients' motivation to initiate and adhere to antiretroviral regimens, and used in alleviating self-stigma [6].

The introduction and subsequent optimization of combination antiretroviral therapy (cART) greatly improved the prognosis of AIDS and the occurrence of other associated events. But, ART drug discontinuation in some people living with HIV (PLHIV) as a result of adverse drug reaction is one of the factors that leads to bad outcome including early mortality. The type of first line ART drug used also determines ART drug tolerability and discontinuation. In some studies, high rate of drug discontinuation was reported among individuals on Dolutegravir (DTG) based regimen as a result of its gastrointestinal or neuropsychological toxicities. From studies conducted in Italy, the rate of DTG based regimen discontinuation was reported between 9.2% and 44% [7, 8]. This drug interruption results in the decline of immunity so that the patient will exposed to opportunistic infections and preventable early mortality.

Antiretroviral therapy significantly decreases mortality overall, but death rates are also highest in the first year, especially, first three months after ART initiation. Opportunistic infections (OIs), immune reconstitution inflammatory syndrome (IRIS), as well as early adverse

drug reactions especially in the first three months of treatment, may be developed. Death rates after initiation of ART are more common when patients start ART with advanced HIV disease stage, with existing comorbidities, severely low baseline clinical and immunological markers like low hemoglobin, low Cluster of Differentiation Four (CD4) cell counts [9, 10], or severely malnourished status. Poor adherence in the first year of ART initiation is also another risk factor that can result in early mortality [4, 11–13].

Patient's Access to ART both in developed and developing countries is improved so that death rates became subsequently decreased. However, the death rate of HIV infected patients in resource-limited countries is still high as compared to those patients from the developed world. This is true especially in the early months after ART initiation. Therefore, in order to reduce such early mortality, identification of possible risk factors is important [14].

Even though survival in HIV-infected patients has improved with Highly Active Antiretroviral Therapy (HAART), still there is a high risk of mortality due to late presentation which is defined as the coming of HIV-infected individuals for care with a CD4 count below 350 cells/ mL [15, 16]. The longer the therapy is delayed, the more the CD4 cell count drops and the poorer the patient outcome. Additionally, delayed presentation for care results in delayed treatment, higher medical costs, and an increased risk of disease transmission by infected individuals who were unaware of their infection status [17, 18].

Different studies identified different factors that can determine the occurrence of early mortality among HIV-infected patients on ART. From these, male sex, advanced age, Tuberculosis (TB) co-infection, low baseline CD4 count, low body mass index, pre-ART viral load, advanced disease stage, bedridden or ambulatory functional status, and low baseline hemoglobin level were stated as independent predictors of mortality. Similarly, factors related to treatment like ART regimen change and poor ART adherence were commonly identified as predictors of early mortality among patients on ART [13, 19–22].

A retrospective cohort study conducted in Harar and Debre-Markos, Ethiopia, showed that mortality of patients on HAART was high. It stated that majority of the death occurred in the first three months of ART initiation. World Health Organization (WHO) clinical stage III and IV, CD4 counts less than 50 cells/µl, not taking Cotrimoxazole Prophylaxis Treatment at the baseline were the identified predictors for early mortality after ART initiation [14, 23].

There is no study conducted in the current study area especially after 2014 after which HIV treatment guidelines were changed. Therefore, this study aimed to assess incidence of the mortality rate and its predictors at Debre-Markos referral hospital.

## Method and materials

### Study design, setting and period

An institution based retrospective follow up study was conducted at Debre-Markos Referral Hospital which is found in Debre Markos town, northwest Ethiopia. Since the start of HIV care, the hospital has providing ART service for around 6, 350 patients and 5, 839 were adults. Of this, 1264 HIV infected adults were enrolled in to ART Clinic between Jan 1, 2014 and December 31, 2018.

### Inclusion and exclusion criteria

All adults age 15 years and above who were newly started ART at Debre-Markos Referral Hospital from Jan 1, 2014 to December 31, 2018, were included in this study. Patients with transferred in record information were excluded.

## Sample size and sampling procedure

Sample size was estimated by using single population proportion formula through EPI INFO statistical package version 7.2.2.6 with the assumption of 95% level of confidence, proportion (P) of 12.5% [24], 3% marginal error, and Power 80%. With these assumptions calculated sample size became 467 and by considering 10% expected incomplete record, the final sample size was 514. Between January 1, 2014, and December 31, 2018, about 1264 adult patients were enrolled to ART, and 1117 fulfill the inclusion criteria. The patients' chart number was taken from the electronic data base of the Hospital, and 514 charts were selected through a computer generated random number.

## Operational definition

**Death or mortality.** Defined as any recorded AIDS-related deaths other than deaths due to accidents like car accident, bullet injury, and any suicidal act.

**Functional status of patients.** Classified according to the WHO criteria as working(W): capably of going out of home and do routine activities including the daily work; Ambulatory (A): capable of self-care and going to the toilet unsupported; Bed-ridden (B): cannot go even to the toilet unsupported [25].

**Baseline WHO clinical stages.** Taken from chart record at enrolment to ART based on WHO classification criteria for HIV/AIDS patients and labeled as stage I to IV.

**Anemia.** In this study anemia is defined as anemic or not-anemic based on WHO criteria: i.e. Hemoglobin concentration <12 g/dl for females and <13 g/dl for males [26].

**ART adherence.** In this study adherence level to ART drug is classified as, Good ($\geq$95% adherence or missing 1 out of 30 doses or missing 2 out of 60 doses), Fair (85–94% adherence or missing 2–4 out of 30 doses or missing 4–9 out of 60 doses), Poor (less than 85% or missing $\geq$5 doses of 30 doses or $\geq$ 10 dose out of 60 doses) [27].

**Past opportunistic infection.** The presence of opportunistic infection/s after HIV diagnosis and before ART start.

**Past CPT treatment.** A patient took CPT after HIV diagnosis and before ART start to prevent the occurrence of OIs.

**Past Isoniazid (INH) prophylaxis.** A patient took INH prophylaxis after HIV diagnosis and before ART start to prevent the occurrence of tuberculosis for those who were ruled out from Tuberculosis disease based on the record on the patients' chart, irrespective of the type of laboratory performed.

**Past TB treatment history.** A patient took anti-tuberculosis treatment before the start of ART.

## Data collection tools and procedures

In order to extract data from the patients' chart, a tool was developed from HIV/AIDS care monitoring and evaluation sheet. In preparing of the tool, forms used for laboratory request and ART intake were considered and incorporated. Prior to data collection, training was given for data collectors regarding the tool and the way they extract the data from the chart. Three trained nurses who had experience on HIV care were recruited for data collection. By using a computer generated simple random sampling technique, patient charts from which actual data taken were selected with the help of medical record number (MRN). Patient's chart was picked up from the chart room using MRN and then data were extracted from the patient's medical charts by using the tool. Common code was given for each selected chart after data was extracted so that there was no chance of recollection of data from a similar chart.

## Data quality control

Pretested data extraction tool was used to maintain data quality. Quality also maintained by extracting data using trained nurses and close monitoring of the procedure by the supervisor. Data clerks were involved upon the selection of patients' chart from the computer as well as from the chart room. Before returning of the chart to the shelf, completeness of data extraction tool was checked and necessary correction was made.

## Data processing and analysis

After the data were extracted from the chart, it was first checked for consistency and completeness. After that it was coded and entered to EPI INFO version 7.2.2.6 and exported to STATA version 14.0 for analysis. A statistical summary was applied to describe socio demographic, clinical and follow up variables of the study. Incidence of death within the follow-up period was calculated and expressed as per 100 person years. Kaplan-Meier was used to describe survival of patients within the follow-up period. Model fitness was checked by using Schoenfield residual test **(p-value = 0.1228)** and goodness of fit was checked by using Cox-Snell residual test. Bi-variable analysis was checked for each variable and those variables with p-value < 0.2 was entered to multivariable Cox-proportional hazard model to identify predictors of early mortality of patients after ART initiation. 95% CI of hazard ratio was computed and variables having p-value < 0.05 in the multivariate Cox proportional hazards model were considered as significant predictors of mortality within the first year.

## Ethical consideration

Ethical clearance was obtained from the Institutional Review Board of the University of Gondar. Upon the ethical clearance, a letter of cooperation was obtained from the school of nursing to collect data. Prior to extracting the data from the patients' chart, permission was obtained from Debre-Markos Hospital Medical director and ART focal person. Confidentiality was maintained and the data were fully anonymized.

## Results

### Baseline socio demographic characteristics of study participants

Out of all patients enrolled to the ART Clinic from January 1, 2014, to December 31, 2018, 514 charts were selected and reviewed based on the inclusion criteria. From these, 494 were included in the analysis and the remaining 20 (3.89%) charts were excluded due to data incompleteness.

Off all patient' charts included in the analysis, 303(61.34%) were females and the median age was 33 years (Inter Quartile Range (IQR): 27–40 years). More than half (54.05%) of the total subjects got married and around 34.62% had no formal education. Majority (78.54%) of the patients were urban dwellers and a total of 450 (91.09%) patients had disclosed their HIV status (Table 1).

### Mortality rate within the first year of ART initiation stratified by socio-demographic characteristics

In one year follow-up period, 494 subjects were followed for a total of 398.37 PY observations. The median follow-up period was 12 months (IQR: 7.23–12) with minimum and maximum follow-ups of 0.93 and 12 months respectively.

**Table 1. Mortality rate within one year follow-up period stratified by socio-demographic characteristics of adults on ART at Debre-Markos Referral Hospital from January 1, 2014 to December 31, 2018 (n = 494).**

| Characteristics | Frequency (%) | Person year | Died | Censored | Death incidence per 100 PY (95% CI) |
|---|---|---|---|---|---|
| Sex | | | | | |
| Male | 191 (38.66) | 147.06 | 24 | 167 | 16.32 (10.94, 24.35) |
| Female | 303 (61.34) | 251.32 | 30 | 273 | 11.94 (8.35, 17.07) |
| Age in years | | | | | |
| 15–24 | 61 (12.35) | 51.94 | 4 | 57 | 7.70 (2.89, 20.52) |
| 25–34 | 200 (40.48) | 160.79 | 21 | 179 | 13.06 (8.52, 20.03) |
| 35–44 | 160 (32.39) | 126.59 | 19 | 141 | 15.01 (9.57, 23.53) |
| $\geq$45 | 73 (14.78) | 59.05 | 10 | 63 | 16.94 (9.11, 31.48) |
| Religion | | | | | |
| Orthodox | 472 (95.55) | 381.24 | 51 | 421 | 13.38 (10.17, 17.60) |
| Muslim | 20 (4.05) | 15.99 | 3 | 17 | 18.75 (6.05, 58.14) |
| Others [b] | 2 (0.40) | 1.13 | 0 | 2 | 0 |
| Marital status | | | | | |
| Single | 62 (12.55) | 42.59 | 9 | 53 | 21.13 (10.99, 40.61) |
| Married | 267 (54.05) | 216.22 | 24 | 243 | 11.09 (7.44, 16.56) |
| Divorced/separated | 129 (26.11) | 110.57 | 16 | 113 | 14.47 (8.87, 23.62) |
| Widowed | 36 (7.29) | 28.99 | 5 | 31 | 17.25 (7.18, 41.44) |
| Educational level | | | | | |
| No education | 171 (34.62) | 138.67 | 17 | 154 | 12.26 (7.62, 19.72) |
| Primary | 113 (22.87) | 92.37 | 8 | 105 | 8.66 (4.33, 17.32) |
| Secondary+ | 210 (42.51) | 167.33 | 29 | 181 | 17.33 (12.04, 24.94) |
| Residence | | | | | |
| Rural | 106 (21.46) | 80.87 | 16 | 90 | 19.79 (12.12, 32.29) |
| Urban | 388 (78.54) | 317.50 | 38 | 350 | 11.97 (8.71, 16.45) |
| Occupation | | | | | |
| Employed | 163 (33.00) | 123.11 | 22 | 141 | 17.87 (11.77, 27.14) |
| Unemployed | 331 (67.00) | 275.26 | 32 | 299 | 11.63 (8.22, 16.44) |
| Household number | | | | | |
| $\leq$ 2 person | 197 (39.88) | 155.57 | 26 | 171 | 16.71 (11.38, 24.55) |
| >2 person | 297 (60.12) | 242.80 | 28 | 269 | 11.53 (7.96, 16.70) |
| HIV disclosure status | | | | | |
| Disclosed | 450 (91.09) | 362.47 | 47 | 403 | 12.97 (9.74, 17.26) |
| Not disclosed | 44 (8.91) | 35.91 | 7 | 37 | 19.49 (9.29, 40.89) |

[b] Protestant and Catholic.

CI: Confidence Interval; PY: Person-Year.

Fifty-four (10.93%; 95% CI: 8.72, 13.62) non-injury related deaths occurred in one year follow-up period with an overall density of 13.56 deaths per 100 PYs (95%CI: 10.38, 17.69). From total deaths that occurred during the follow-up period, more than half (55.56%) were females but the death rate was high in males than females which was 16.32 and 11.94 deaths per 100 PYs, respectively. The highest death rate was observed in the group among subjects age 45 years and above which was 16.94 deaths per 100 PYs. From the cohort, a higher death rate was observed among subjects from a rural area and also it was higher (19.49 deaths per 100 PY) among participants who didn't disclose their HIV status (Table 1).

### Baseline and follow up clinical and immunological characteristics of study participants

Out of 494 study participants with complete information for analysis, more than two-thirds (70.85%) of them had started ART within six months of HIV status confirmation and only 16% had past opportunistic infection. A total of 215 (43.52%) patients were in baseline WHO clinical stage I and the median baseline CD4 count was 314.5 cells/μl (IQR: 147–443). Three hundred and two (61.13%) participants were found to be grouped under the 18.5 to 24.9 kg/m$^2$ category of body mass index (BMI). At baseline, 166 (33.60%) patients were anemic, and also majority of (79.35%) the patients had working functional status at baseline.

From those patient charts included in the analysis, about two-third (65%) had taken CPT and less than half (39.47%) of the patients took Isoniazid preventive therapy (IPT) during the follow up period. On one year follow-up time, 43 (8.70%) patients on ART developed tuberculosis (TB). At baseline, almost all (94.53%) of the patients start ART with the category of Tenofovir Disoproxil Fumarate-Lamivudine- Efavirenz (TDF-3TC-EFV), and around 29% of patients experience fair or poor ART adherence during the follow up period (Table 2).

### Mortality rate within the first year of ART initiation stratified by baseline and follow-up clinical and immunological characteristics

During the follow-up period more death (21.25 deaths per 100 PY) was observed with patients who started ART six months after HIV confirmation. Under the category of baseline clinical staging, the highest death was observed among those with baseline clinical stage IV (25.9 deaths per 100 PYs). Around 15 deaths per 100 PYs occurred on patients having baseline CD4 cell count less than 100 cells/μl, and also it was 17.35 deaths/100PY on those categorized with baseline BMI of less than 18.5kg/m2. A higher death rate was found in patients who had anemia at baseline (18.89 deaths/100 PY). Similarly, the death rate after ART initiation was high on those patients who had bedridden functional status at baseline (47.96 deaths/100PY). More death rates also recorded among patients who didn't take CPT and IPT during the follow-up period (17.69 and 17.77 deaths/100PY, respectively). A total of 43 patients had developed tuberculosis within one year follow-up period, of these seven patients had died (31.59 deaths/100 PY). During the follow-up period, eleven deaths (13.95 deaths/100 PY) were recorded with those who had poor ART adherence (Table 2).

### Over all Kaplan-Meier survival curve from ART initiation to death

The probability of surviving from ART initiation to death within one-year follow-up period was estimated. So, the survival probability for the total cohort at the end of 3 months was 0.96 (95% CI: 0.93–0.97); at the end of 6 months was 0.92 (95% CI: 0.89–0.94); at the end of 9 months was 0.89 (95% CI: 0.86–0.92); and that of the probability of surviving at the end of follow-up was 0.88 (95% CI: 0.84–0.91) (Fig 1).

### Predictors of mortality within one year of ART initiation

In the bivariable Cox-regression analysis, duration from HIV status confirmation to ART initiation (pre-ART duration), age at enrolment to ART, level of education, residence, employment status, household number, WHO clinical staging, body mass index, functional status, anemia status at baseline, took IPT, took CPT during follow up, and TB co-infection during the follow up were found to be predictors of death within one year of ART initiation at a P-value of less than 0.2. So, these variables were also selected for multivariable Cox-regression analysis. From these, pre-ART duration, residence, functional status at baseline, and took CPT during

**Table 2. Mortality rate stratified by baseline and follow up clinical and immunological characteristics of adults on ART at Debre-Markos Referral Hospital from January 1, 2014 to December 31, 2018 (n = 494).**

| Characteristics | Frequency (%) | Person year | Died | Censored | Death/100 PY (95% CI) |
|---|---|---|---|---|---|
| Pre-ART duration | | | | | |
| < 6 month | 350 (70.85) | 280.73 | 29 | 321 | 10.33 (7.18, 14.87) |
| ≥ 6 month | 144 (29.15) | 117.64 | 25 | 119 | 21.25 (14.36, 31.45) |
| Past OI | | | | | |
| Yes | 79 (15.99) | 65.63 | 7 | 72 | 10.67 (5.09, 22.37) |
| No | 415 (84.01) | 332.75 | 47 | 368 | 14.12 (10.61, 18.79) |
| Past CPT treatment | | | | | |
| Yes | 93 (18.83) | 82.62 | 7 | 86 | 8.47 (4.04, 17.77) |
| No | 401 (81.17) | 315.75 | 47 | 354 | 14.89 (11.18, 19.81) |
| Past INH prophylaxis | | | | | |
| Yes | 15 (3.04) | 14.16 | 0 | 15 | 0 |
| No | 479 (96.96) | 384.21 | 54 | 425 | 14.05 (10.76, 18.35) |
| Past TB treatment history | | | | | |
| Yes | 480 (97.17) | 12.91 | 0 | 14 | 0 |
| No | 14 (2.83) | 385.46 | 54 | 426 | 14.01 (10.73, 18.29) |
| Baseline clinical staging | | | | | |
| I | 215 (43.52) | 172.29 | 19 | 196 | 11.03 (7.03, 17.29) |
| II | 124 (25.10) | 108.04 | 14 | 110 | 12.96 (7.67, 21.88) |
| III | 126 (25.51) | 98.75 | 16 | 110 | 16.20 (9.93, 26.45) |
| IV | 29 (5.87) | 19.31 | 5 | 24 | 25.90 (10.78, 62.23) |
| Baseline CD4 count (cells/μl) | | | | | |
| <100 | 89 (18.02) | 67.45 | 10 | 79 | 14.83 (7.98, 27.56) |
| 100–199 | 74 (14.98) | 61.99 | 7 | 67 | 11.29 (5.38, 23.69) |
| 200–349 | 124 (25.10) | 102.79 | 14 | 110 | 13.62 (8.07, 22.99) |
| ≥ 350 | 207 (41.90) | 166.15 | 23 | 184 | 13.84 (9.19, 20.83) |
| Baseline BMI | | | | | |
| < 18.5 | 137 (27.73) | 97.99 | 17 | 120 | 17.35 (10.78, 27.91) |
| 18.5–24.9 | 302 (61.13) | 254.10 | 33 | 269 | 12.99 (9.23, 18.27) |
| >24.9 | 55 (11.13) | 46.27 | 4 | 51 | 8.64 (3.24, 23.03) |
| Baseline Anemia | | | | | |
| Yes | 166 (33.60) | 121.69 | 23 | 143 | 18.89 (12.56, 28.44) |
| No | 328 (66.40) | 276.67 | 31 | 297 | 11.21 (7.88, 15.93) |
| Baseline functional status | | | | | |
| Working | 392 (79.35) | 329.61 | 36 | 356 | 10.92 (7.88, 15.14) |
| Ambulatory | 94 (19.03) | 64.59 | 16 | 78 | 24.77 (15.17, 40.43) |
| Bedridden | 8 (1.62) | 4.17 | 2 | 6 | 47.96 (11.99, 191.7) |
| Took CPT at follow up | | | | | |
| Yes | 321 (64.98) | 274.00 | 32 | 289 | 11.68 (8.26, 16.51)) |
| No | 173 (35.02) | 124.37 | 22 | 151 | 17.69 (11.65, 26.87) |
| Took IPT at follow up | | | | | |
| Yes | 195 (39.47) | 167.59 | 13 | 182 | 7.76 (4.50, 13.36) |
| No | 299 (60.53) | 230.78 | 41 | 258 | 17.77 (13.08, 24.13) |
| TB/HIV co-infection | | | | | |
| Yes | 43 (8.70) | 22.16 | 7 | 36 | 31.59 (15.06, 66.26) |
| No | 451 (91.30) | 376.21 | 47 | 404 | 12.49 (9.39, 16.63) |
| Initial ART regimen | | | | | |

*(Continued)*

**Table 2.** (Continued)

| Characteristics | Frequency (%) | Person year | Died | Censored | Death/100 PY (95% CI) |
|---|---|---|---|---|---|
| TDF/3TC/EFV | 467 (94.53) | 376.65 | 50 | 417 | 13.27 (10.06, 17.52) |
| Others [c] | 27 (5.47) | 21.72 | 4 | 23 | 18.41 (6.91, 49.06) |
| ART Adherence | | | | | |
| Good | 350 (70.85) | 293.07 | 35 | 315 | 11.94 (8.57, 16.63) |
| Fair | 40 (8.10) | 26.50 | 8 | 32 | 30.19 (15.09, 60.36) |
| Poor | 104 (21.05) | 78.79 | 11 | 93 | 13.95 (7.73, 25.21) |

[c] AZT/3TC/EFV, TDF/3TC/NVP, AZT/3TC/LPV/r, AZT/3TC/ATV/r, TDF/3TC/ATV/r.

3TC: Lamivudine; ABC: Abacavir; ART: Antiretroviral Therapy; AZT: Zidovudine; BMI: Body Mass Index; CD4: Cluster of Differentiation four; CI: Confidence
Interval; CPT: Cotrimoxazole Preventive Therapy; EFV: Efavirenz; HIV: Human Immunodeficiency Virus; INH: Isoniazid; IPT: Isoniazid Preventive Therapy; LPV:
Lopinavir; NVP: Nevirapine; OI: Opportunistic Infection; PY: Person Year; TDF: Tenofovir Disoproxil Fumarate; TB: Tuberculosis.

follow-up were found to be statistically significant predictors of the death rate among adults within the first year of ART initiation at a P-value of less than 0.05. But, variables like presence of opportunistic infections at the moment of diagnosis and CD4 cells/count were not included in the Cox model because the p-value at bi-variable analysis was greater than 0.2 (Table 3).

Assumptions of Cox-proportional hazard were also checked for these variables and they didn't violate the assumption (Figs 2–5).

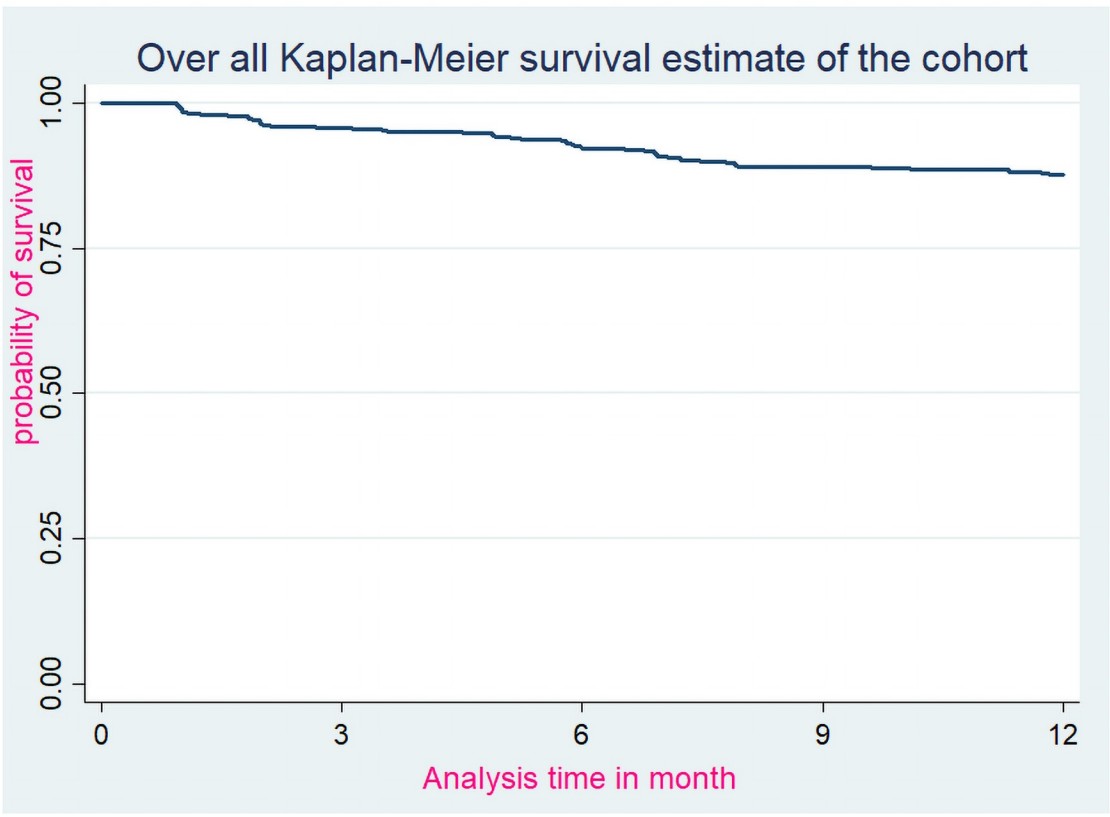

**Fig 1. Over all Kaplan-Meier survival curve from ART initiation to death within one year follow-up period among adults on ART at Debre-Markos Referral Hospital from Jan 1, 2014 to Dec 31, 2018.**

**Table 3. Cox regression analysis of predictors of mortality within one year of ART initiation among adults on ART at Debre-Markos Referral Hospital; Jan 1, 2014 to Dec 31, 2018 (n = 494).**

| Characteristics | Outcome | | CHR (p-value) | AHR (95% CI) | P-value |
|---|---|---|---|---|---|
| | **Died** | **Censored** | | | |
| Patient's Age in years | | | | | |
| 15–24 | 4 | 57 | 1.00 | 1.00 | |
| 25–34 | 21 | 179 | 1.68 (0.340) | 1.91(0.64, 5.69) | 0.245 |
| 35–44 | 19 | 141 | 1.93 (0.231) | 2.11(0.69, 6.41) | 0.186 |
| $\geq$ 45 | 10 | 63 | 2.16 (0.192) | 3.05 (0.89, 10.46) | 0.077 |
| Level of education | | | | | |
| No education | 17 | 154 | 0.71 (0.256) | 0.54 (0.29, 1.03) | 0.062 |
| $1^0$ education | 8 | 105 | 0.50 (0.083) | 0.46 (0.20, 1.03) | 0.058 |
| $2^0$ & above | 29 | 181 | 1.00 | 1.00 | |
| Patient's Occupation | | | | | |
| Employed | 22 | 141 | 1.00 | 1.00 | |
| Un employed | 32 | 299 | 0.67 (0.144) | 0.69 (0.39, 1.23) | 0.212 |
| Patient's Residence | | | | | |
| Rural | 16 | 90 | 1.65 (0.094) | 1.97 (1.05, 3.71) | **0.035**[*] |
| Urban | 38 | 350 | 1.00 | 1.00 | |
| Patient Household number | | | | | |
| $\leq$ 2 person | 26 | 171 | 1.00 | 1.00 | |
| >2 persons | 28 | 269 | 0.69 (0.178) | 0.55 (0.29, 1.01) | 0.051 |
| Pre-ART duration | | | | | |
| < 6 months | 29 | 321 | 1.00 | 1.00 | |
| $\geq$ 6 months | 25 | 119 | 2.07 (0.008) | 2.17 (1.24, 3.79) | **0.006**[*] |
| Baseline Anemia | | | | | |
| Yes | 23 | 143 | 1.66 (0.066) | 1.19 (0.60, 2.34) | 0.617 |
| No | 31 | 297 | 1.00 | 1.00 | |
| Body mass index | | | | | |
| <18.5 kg/m$^2$ | 17 | 120 | 1.29 (0.182) | 0.81 (0.42, 1.56) | 0.526 |
| 18.5–24.9 kg/m$^2$ | 33 | 269 | 1.00 | 1.00 | |
| > 24.9 kg/m$^2$ | 4 | 51 | 0.66 (0.440) | 0.56 (0.19, 1.64) | 0.292 |
| Baseline clinical Staging | | | | | |
| I/II | 33 | 306 | 1.00 | 1.00 | |
| III | 16 | 110 | 1.36 (0.308) | 1.02 (0.49, 2.09) | 0.953 |
| IV | 5 | 24 | 2.09 (0.124) | 1.31 (0.43, 4.02) | 0.634 |
| Baseline functional status | | | | | |
| Working | 36 | 356 | 1.00 | 1.00 | |
| Ambulatory/bedridden | 18 | 84 | 2.31 (0.004) | 2.18 (1.01, 4.74) | **0.049**[*] |
| Took IPT during follow up | | | | | |
| Yes | 13 | 182 | 1.00 | 1.00 | |
| No | 41 | 258 | 2.28 (0.010) | 1.87 (0.97, 3.59) | 0.060 |
| Took CPT at follow up | | | | | |
| Yes | 32 | 289 | 1.00 | 1.00 | |
| No | 22 | 151 | 1.48 (0.160) | 1.88 (1.04, 3.41) | **0.038**[*] |
| TB Co-infection | | | | | |
| Yes | 7 | 36 | 2.35 (0.036) | 1.73 (0.66, 4.55) | 0.264 |

(*Continued*)

**Table 3.** (Continued)

| Characteristics | Outcome | | CHR (p-value) | AHR (95% CI) | P-value |
|---|---|---|---|---|---|
| | Died | Censored | | | |
| No | 47 | 404 | 1.00 | 1.00 | |

*shows statistically significant variables at p-value <0.05.

AHR: Adjusted Hazard Ratio; ART: Antiretroviral Therapy; CHR: Crude Hazard Ratio; CI: Confidence Interval; CPT: Cotrimoxazole Preventive Therapy; IPT: Isoniazid Prophylactic Therapy; TB: Tuberculosis

## Discussion

In this study, the overall incidence of mortality within a year of ART initiation and its predictors were identified. Overall, 10.93% (95% CI: 8.72–13.62) of study participants were known to have died within a year of ART initiation. This is in line with studies conducted in Tigray (13.5%) and Somalia (9.8%), Ethiopia [24, 28], and also study in India with 11.9% [29]. This similarity with Tigray and Somalia, Ethiopia, could be that they used similar HIV care and treatment guideline. Likewise, Indians and Ethiopians practice almost similar lifestyles which might contribute to this HIV-related result.

The cumulative incidence of death during the follow-up was 13.56 deaths per 100 PY (95% CI: 10.38–17.69). This is in line with a study conducted from South Africa which showed the mortality rate of 13 deaths per 100 PY [30]. But, the death rate of the current study is higher

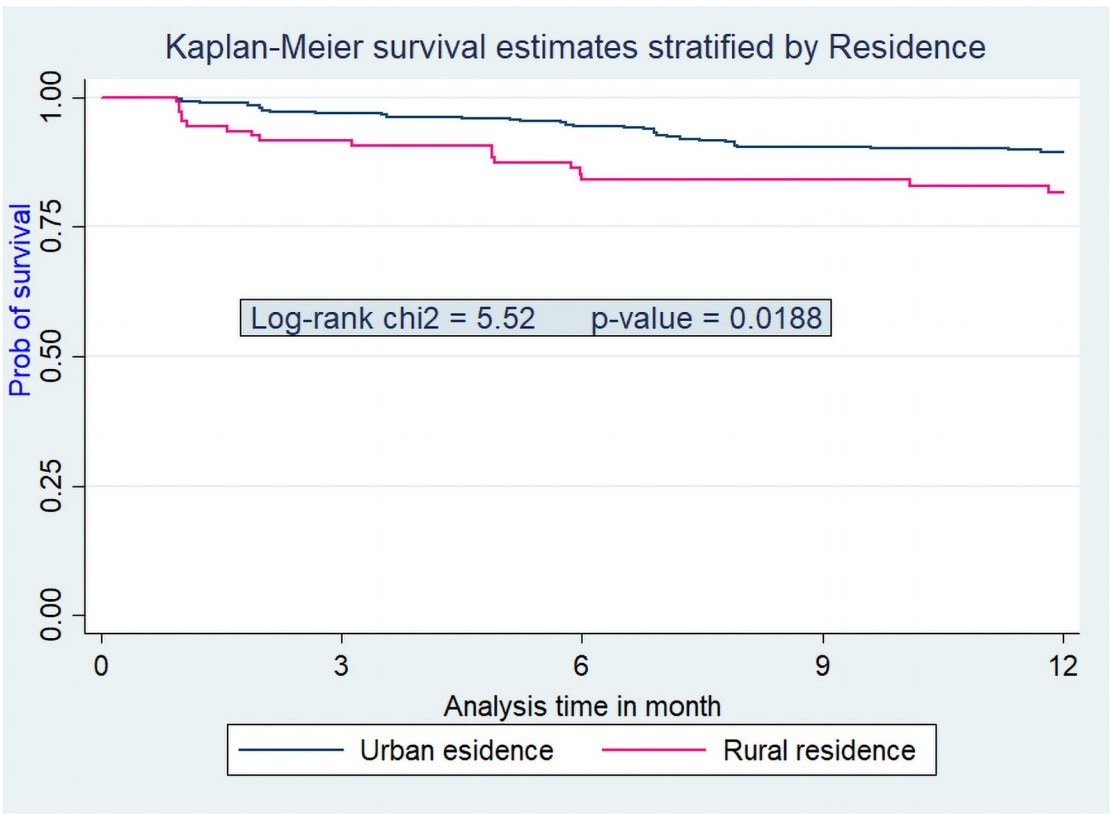

**Fig 2. Kaplan Meier survival curve from ART initiation to death within one year stratified by residence among adults on ART at Debre-Markos Referral Hospital from Jan 1, 2014, to Dec 31, 2018.**

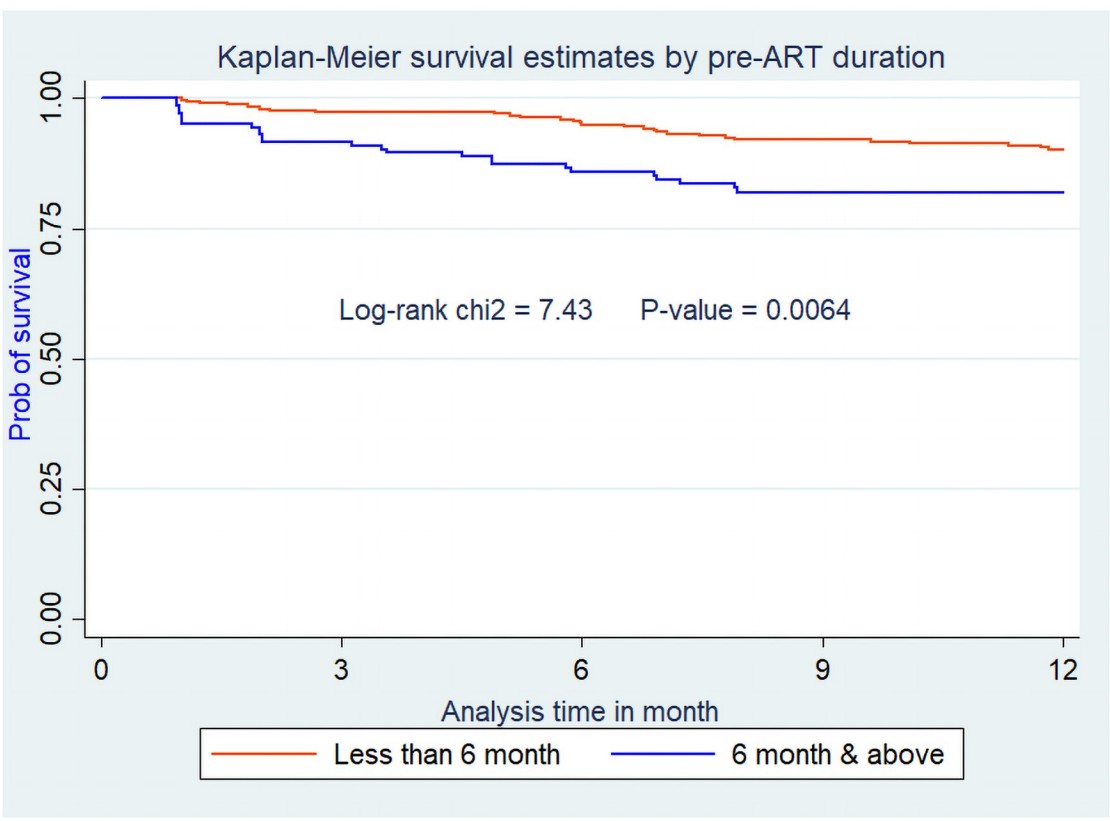

**Fig 3. Kaplan Meier survival curve from ART initiation to death within one year stratified by pre-ART duration among adults on ART at Debre-Markos Referral Hospital from Jan 1, 2014, to Dec 31, 2018.**

than similar studies conducted in Uganda, India, and Korea [11, 19, 22]. This higher rate of death in the current study might be due to difference in the study setting, the difference in time of the study, and the lower sample size used in the current study. The lower economical level of the current study setting than Indian and Korean Gross Domestic Product (GDP) also contribute for a higher mortality rate of HIV-infected patients.

In the current study, more death rate was observed within the first three months after ART initiation which was 17.77 deaths per 100 PY (95% CI: 11.59, 27.26). This is congruent with a study from South Africa [30]. This might be that patients within the first three months of ART initiation may face many difficulties like IRIS, adverse effects of HAART, and high susceptibility to OIs which results in early mortality.

In this study, factors that predict early mortality within the first year after ART initiation were identified. After applying multivariable analysis, four variables including patient residence, pre-ART duration, functional status at the baseline and status of CPT usage while on ART were found to be independently significant predictors of mortality within the first year.

This study revealed that patients from rural residence were about two times (AHR = 1.97; 95% CI: 1.05, 3.71) at higher risk of death within the first year after ART initiation as compared to patients from urban residence. The possible justification for this finding could be patients from rural residence are more likely to be far from HIV care settings and are prone to missing of appointments for their next schedules. Also these patients are less aware about side effects of ART drugs that they took for treatment; so that they may discontinue the treatment. By these

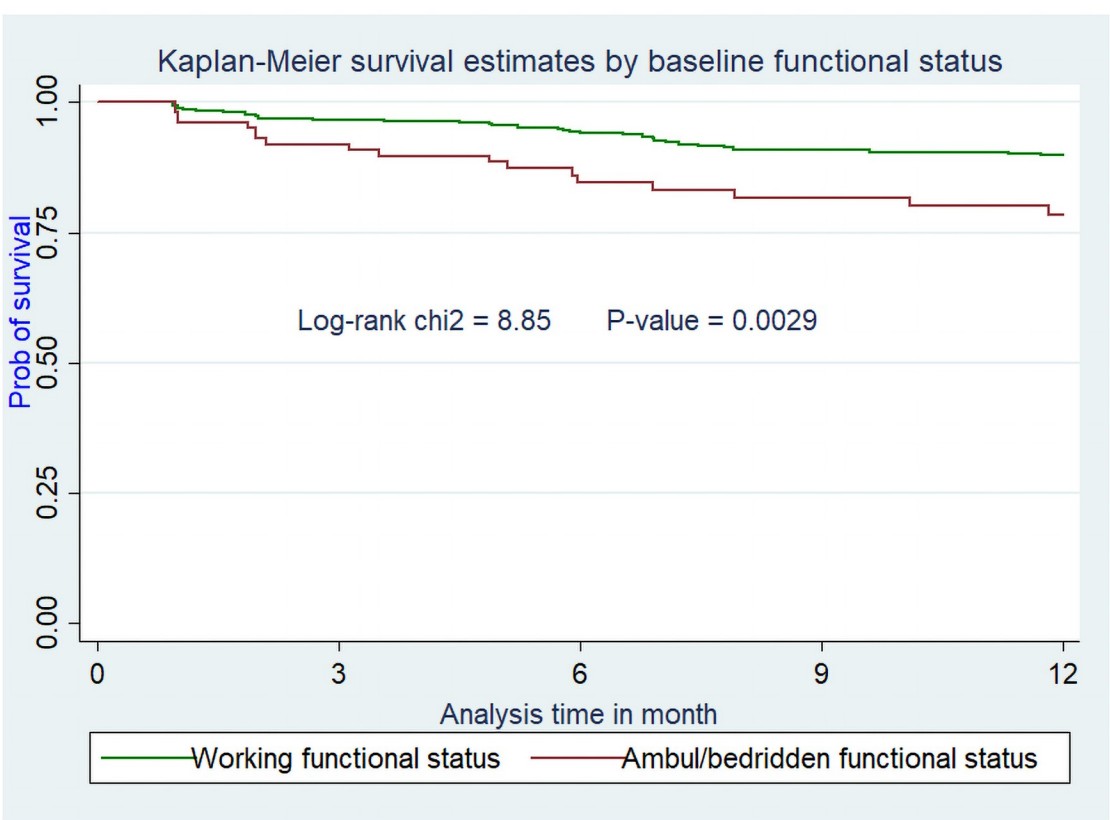

**Fig 4. Kaplan Meier survival curve from ART initiation to death within one year stratified by baseline functional status among adults on ART at Debre-Markos Referral Hospital from Jan 1, 2014, to Dec 31, 2018.**

and other factors patients from rural residence tends to be less adhere to ART drugs [31–33] and this non-adherence can contribute to early mortality of HIV infected patients.

Similarly, HIV infected patients who started ART six months and above after HIV status confirmation were 2.17 times (AHR = 2.17; 95% CI: 1.24, 3.79) at higher risk of death within the first year than patients who started ART early. This is supported by similar study conducted in Uganda at 2012 [11]. This might be due to the fact that as the pre-ART duration became more prolonged, the disease progresses most probably to the advanced HIV disease status. This HIV disease advancement in turn results in an increased risk of death even after starting ART [34].

Patients with baseline ambulatory or bedridden functional status were about two fold at higher risk of death within the first year of HAART as compared to patients with working functional status at enrolment to ART (AHR = 2.18; 95% CI: 1.01, 4.74). This finding is agreed with other studies conducted in different parts of Ethiopia at different time periods, India, and China [20, 28, 29, 35–39]. This is due to the fact that ambulatory or bedridden functional status is an indicator of low immune status. When the immune status of HIV infected patients became low, they tend to coerced for inability to cope with the disease and high risk of death due to OIs [40].

WHO guidelines on Cotrimoxazole prophylaxis in resource-limited settings recommended the use of this prophylaxis as an integral component of HIV care to prevent OIs and related mortality [4]. But, different studies stated that, still many patients were dying as a result of not taking CPT. In the current study, patients who didn't take CPT during the follow up were 1.88

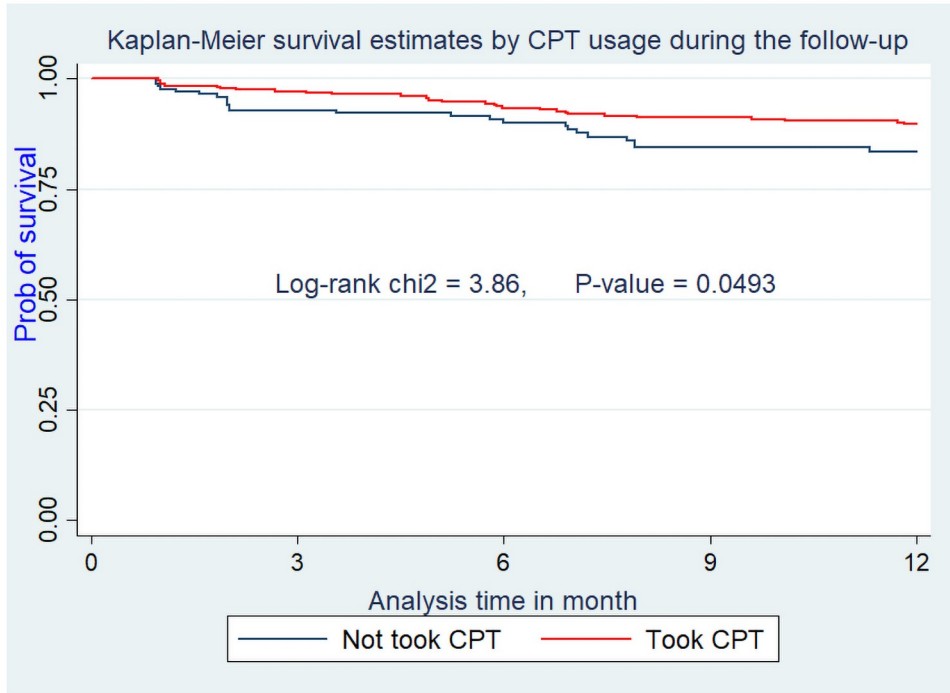

**Fig 5. Kaplan Meier survival curve from ART initiation to death within one year stratified by CPT usage among adults on ART at Debre-Markos Referral Hospital from Jan 1, 2014, to Dec 31, 2018.**

times at higher risk of death as compared to those who took CPT (AHR = 1.88; 95% CI: 1.04–3.41). This is supported by other studies conducted in Debre-Markos and Harar, Ethiopia [14, 23]. This higher risk of death among CPT non-users might be due to the fact that, as HIV infected patients didn't use CPT they are less benefited from the implication of Cotrimoxazole prophylaxis. CPT significantly reduces the occurrence of opportunistic infections which can be a risk factors for early mortality among patients on HAART [4].

## Conclusion

Mortality rate of HIV patients in the first year of ART initiation is still high in the study area.

Rural residence, longer pre-ART duration, not took CPT, and ambulatory/bedridden functional status at enrolment were found to be independent predictors of mortality within the first year of ART initiation. Hence, giving special attention for patients from rural area and provision of CPT are crucial to reduce mortality. If possible, initiation of ART sooner after status confirmed is crucial especially for those patients who are more at risk of developing low immunity.

## Limitation

Because of its retrospective nature of the study few important variables like smoking, chat chewing, and alcohol drinking were not included because these variables were not recorded on the chart. Additionally, HIV-RNA related data was not incorporated due to lack of this test for majority of patients.

## Supporting information

**S1 Table. Bi-variable Cox regression analysis of predictors of mortality within one year of ART initiation among adults on ART at Debre-Markos Referral Hospital; Jan 1, 2014 to Dec 31, 2018 (n = 494).**
(DOCX)

**S1 Dataset. Anonymized dataset.**
(XLS)

## Acknowledgments

The authors would like to acknowledge the hospital director and data collectors for their collaboration during the data collection. Also, authors' heartfelt thank was goes to University of Gondar for providing ethical clearance.

## Author Contributions

**Conceptualization:** Agazhe Aemro.

**Data curation:** Agazhe Aemro, Mulugeta Wassie, Basazinew Chekol.

**Formal analysis:** Agazhe Aemro, Mulugeta Wassie.

**Investigation:** Agazhe Aemro.

**Methodology:** Agazhe Aemro.

**Project administration:** Agazhe Aemro.

**Resources:** Agazhe Aemro, Mulugeta Wassie, Basazinew Chekol.

**Software:** Agazhe Aemro.

**Supervision:** Agazhe Aemro, Mulugeta Wassie, Basazinew Chekol.

**Validation:** Agazhe Aemro, Mulugeta Wassie.

**Visualization:** Agazhe Aemro, Mulugeta Wassie, Basazinew Chekol.

**Writing – original draft:** Agazhe Aemro.

**Writing – review & editing:** Agazhe Aemro, Mulugeta Wassie, Basazinew Chekol.

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
