## [Decision Letter · Decision Letter 0]

17 Dec 2020

PONE-D-20-34000

Incidence and predictors of mortality within the first year of antiretroviral therapy initiation at Debre-Markos referral hospital, Northwest Ethiopia: A retrospective follow up study

PLOS ONE

Dear Dr. Aemro,

Thank you for submitting your manuscript to PLOS ONE. After careful consideration, we feel that it has merit but does not fully meet PLOS ONE’s publication criteria as it currently stands. Therefore, we invite you to submit a revised version of the manuscript that addresses the points raised during the review process.

We look forward to receiving your revised manuscript.

Kind regards,

Giordano Madeddu

Academic Editor

PLOS ONE

Journal Requirements:

4. Please ensure you have thoroughly discussed any potential limitations of this study within the Discussion section, including the potential impact of confounding factors.

7. We note you have included a table to which you do not refer in the text of your manuscript. Please ensure that you refer to Table 4. in your text; if accepted, production will need this reference to link the reader to the Table.

Reviewers' comments:

Reviewer's Responses to Questions

**Comments to the Author**

1. Is the manuscript technically sound, and do the data support the conclusions?

Reviewer #1: Yes

Reviewer #2: Yes

2. Has the statistical analysis been performed appropriately and rigorously? 

Reviewer #1: Yes

Reviewer #2: Yes

3. Have the authors made all data underlying the findings in their manuscript fully available?

Reviewer #1: Yes

Reviewer #2: Yes

4. Is the manuscript presented in an intelligible fashion and written in standard English?

Reviewer #1: Yes

Reviewer #2: No

5. Review Comments to the Author

Reviewer #1: The aims of the study are very interesting and appealing. Indeed, the authors esplored the impact of a large number of variables on the clinical outcomes.

In my opinion, the paper could be more clear and easy to read if results are summarized better, avoiding to duplicate results in table and text. They need to report a valid legend for the KM figures and log Rank p results.

Table 4 is outlined.

Reviewer #2: Agazhe Aemr et al. in their study showed the incidence incidence and predictors of mortality in naive people with HIV starting an antiretroviral treatment. Many issues are present.

General comment:

Abbreviations should be written entirely in the first appearance both in abstract and in full text.

Many typo and grammatical mistakes are present in the manuscript. I suggest to include an English mother tongue revisor to check the manuscript.

Introduction

- The authors wrote "But HIV-infected individuals still hesitates to start the treatment early which results in progression of the disease to the advanced stage. This intern results in risk of early mortality for patients with HIV". Probably the hesitation is due to cultural behavior. I suggest specifying that this sentence concern the African or Ethiopian situation.

About the introduction, I suggest adding a part about the efficacy and tolerability of the new drugs also in the advanced HIV-infected naïve patients. I suggest you read these papers that you could also cite in the introduction: https://doi.org/10.1016/j.antiviral.2019.104552, https://doi.org/10.3390/jcm8122062, https://doi.org/10.2147/IDR.S260449, https://doi.org/10.1097/QAD.0000000000001357

Method

The methods section is quite long. I suggest deleting the geographical part; although it is very interesting, it does not add crucial information.

In the results section, the authors wrote about anemia. I suggest adding the definition of anemia in methods.

Results and discussion

The authors wrote, "but the 179 death rate was high in males which is 16.32 deaths per 100 person year." I suggest also adding the number of female deaths per 100 person-years to make clear the difference.

Also they should define what they intend with good, fair, and poor adherence to antiretroviral treatment.

No data about baseline HIV-RNA load is present. I believe that this information is crucial both for deaths and virological failure. If this information is not available, the authors should discuss it in the limit's section.

The authors wrote, "From these, pre-ART duration, residence, functional status at baseline, and took Cotrimoxazole preventive therapy during follow up were found to be statistically significant predictors of the death rate among adults within the first year of ART initiation at a P-value of less than 0.05 (Table 3)". I think the authors mean "table 4". Please check it.

No data about the cause of death are present. I suggest adding them.

The presence of opportunistic infections at the moment of diagnosis is not present in the Cox model. Which was the CHR of having them?

Also, CD4 cells/count does not been considered in the Cox model. Different studies showed how late-presenters had increased mortality. I believe that the authors should discuss it.

Table 1.

- Please recheck the percentages' approximation. The sum of "age in years", and Educational level" is 100.1% and 99.99% respectively.

- 67% of people had a baseline CD4 cell count > 200 cells-mL and 41.9 more than 350. It is not clear why 65% of people took cotrimoxazole and 39.47% isoniazid. People who start isoniazid (how long did they take it?) had a positive tuberculin test?

Table 2

- It is not clear what the authors mean with "Past OI", "Past CPT treatment", "Past INH prophylaxis", and "Past TB treatment history. Do they mean if patients have done the treatments before HIV diagnosis?

- The authors should include in the table a subtitle to explain all abbreviations used in the table.

Table 3.

- Please remove "(continued)", or rename the table as table 2.

- About the initial ART regimens, the authors choose as variable "le" and "others". Please correct it. Furthermore, I believe that it is interesting knowing what other regimens the patients started. In my opinion, this is important to understand why mortality is higher in "others" group.

- The authors should include in the table a subtitle to explain all abbreviations used in the table.

Table 4

- It should be in the results section and not in the discussion.

- The authors should include in the table a subtitle to explain all abbreviations used in the table.

- In table 2 the authors divided the educational level into "no education", "primary", "secondary", and college +". In this table, instead, they divided it into "no education", "10 education", and "20 & above". I suggest using the same division.

Figure 1

- It is not clear what the number between brackets mean. Please comment.

All figures

- In the text, the authors wrote about mortality using "months of follow-up," while in the figures, time is expressed as a fraction of years. It could create confusion in the readers because "0.6" could be interpreted as "6 months", instead of 7.2 months. I suggest modifyin,g the X-axes of the figures

- I suggest performing a Log-rank test to compare two samples' survival distributions in the different KM to see if there is a statistical difference.

6. PLOS authors have the option to publish the peer review history of their article (what does this mean?). If published, this will include your full peer review and any attached files.

Reviewer #1: No

Reviewer #2: No

---

## [Author Response · Author response to Decision Letter 0]

27 Jan 2021

Point by point response to reviewers’ comments

Part I: Editors’ concerns and response by the authors:

1. Editors’ concern:

Please ensure that your manuscript meets PLOS ONE's style requirements, including those for file naming

Authors’ response: 

The authors tried to ensure that the manuscript meets PLOS ONE's style requirements.

2. Editors’ concern:

Please provide additional details regarding participant consent. 

Authors’ response: 

This study was based on chart review so that individual consent from each participant is not important, even it is impossible to get individual consent from each participant. There for, the one who is responsible to give the consent for the authors to collect data from the chart was medical director of the hospital. So, the medical director gave to the authors a signed consent and we took it and gave to data clerks so that the data collectors could extract data from the charts. Additionally, all data extracted from patient’s chart were fully anonymized.

3. Editors’ concern:

We suggest you thoroughly copyedit your manuscript for language usage, spelling, and grammar.

Authors’ response: 

With repeated reading and with the help of online grammar checker, the authors tried to correct problems regarding to language usage, spelling, and grammar.

4. Editors’ concern:

Please ensure you have thoroughly discussed any potential limitations of this study within the Discussion section, including the potential impact of confounding factors.

Authors’ response: 

Potential limitations of the study were stated in the revised manuscript next to the discussion part. Additionally, confounding factors were reduced by excluding from multi-variable analysis of those variables with p-value of greater than 0.2 at bi-variable analysis. 

5. Editors’ concern:

We note that you have indicated that data from this study are available upon request. PLOS only allows data to be available upon request if there are legal or ethical restrictions on sharing data publicly?

Authors’ response: 

Thank you for your clarification. 

The message that I expressed as “data from this study are available upon request” doesn’t mean there is a restriction of availing the data set. Rather it was to mean that the data will be avail based on your interest or your need. 

a. Editors’ concern:

If there are ethical or legal restrictions on sharing a de-identified data set, please explain them in detail

Authors’ response: 

In this data set, there are no ethical or legal restrictions on sharing a de-identified data set; because, there is no potentially identified or sensitive patient information and it is fully anonymized.

b. As per the request, minimal anonymized data set is uploaded.

6. Editors’ concern:

Your ethics statement should only appear in the Methods section of your manuscript. If your ethics statement is written in any section besides the Methods, please move it to the Methods section and delete it from any other section. Please ensure that your ethics statement is included in your manuscript, as the ethics statement entered into the online submission form will not be published alongside your manuscript

Authors’ response:

Thanks for your information; the ethics statement is incorporated only in the Methods section of the revised manuscript.

7. Editors’ concern:

We note you have included a table to which you do not refer in the text of your manuscript. Please ensure that you refer to Table 4. in your text; if accepted, production will need this reference to link the reader to the Table.

Authors’ response:

Yes, the author accept it and made a correction on the manuscript. Actually there was no any table which the author didn’t refer in the text of the previous manuscript. There were three tables (table 1, table 2, and table 3), of which table 2 was divided in to two pages with a title “table 2 (continued)” and this might resulted in confusion to both the editors and reviewers. By considering this, the authors tried to merge table 2 and put it in a single page of the revised manuscript. 

Part II: Reviewers' comments and response by the authors:

Reviewer #1: The aims of the study are very interesting and appealing. Indeed, the authors explored the impact of a large number of variables on the clinical outcomes. In my opinion, the paper could be more clear and easy to read if results are summarized better, avoiding to duplicate results in table and text. They need to report a valid legend for the KM figures and log Rank p results. Table 4 is outlined.

Authors’ response:

• The authors highlighted main results in the text and tried to summarize in some manner, as per the comment. But, it could be clear that not all findings on the table are incorporated in the text.

• Based on the comment the authors give valid legend for the KM figures.

• Both chi-square and log-rank p-values are incorporated in the revised manuscript. 

• The authors re-organize the tables as table 1, 2, and 3 by removing “table 2 continued”. Also, the tables are cited immediately after the text they tend to be referred. 

Reviewer #2: 

General comment by reviewer#2: 

Abbreviations should be written entirely in the first appearance both in abstract and in full text.

Many typo and grammatical mistakes are present in the manuscript. I suggest to include an English mother tongue revisor to check the manuscript.

Authors’ response:

Thank you for your comment and I tried to consider all the comments; Abbreviations had written entirely in the first appearance and attempt was made to correct grammatical mistakes present in the manuscript with repeated reading and using of online grammar checker (https://www.scribens.com/, https://www.grammarcheck.net/editor/).

Introduction: 

Reviewer concern:

The authors wrote "But HIV-infected individuals still hesitates to start the treatment early which results in progression of the disease to the advanced stage. This intern results in risk of early mortality for patients with HIV". Probably the hesitation is due to cultural behavior. I suggest specifying that this sentence concern the African or Ethiopian situation.

About the introduction, I suggest adding a part about the efficacy and tolerability of the new drugs also in the advanced HIV-infected naïve patients. 

Authors’ response:

• The authors tried to edit the idea regarding the term “hesitation”.

• As per the comment, authors incorporated the effect of Drug discontinuation on the prognosis, on the introduction part. 

Method: 

Reviewer concern:

The methods section is quite long. I suggest deleting the geographical part; although it is very interesting, it does not add crucial information.

In the results section, the authors wrote about anemia. I suggest adding the definition of anemia in methods.

Authors’ Response: 

• Based on the comment, the authors tried to avoid the geographical part in order to shorten the method. 

• As per the comment, the authors included the definition of anemia and ART adherence to the method section of the revised manuscript. 

Result and discussion: 

Reviewer concern: the authors wrote, "But the 179 death rate was high in males which is 16.32 deaths per 100 person year." I suggest also adding the number of female deaths per 100 person-years to make clear the difference

Authors’ Response: 

• We would like to say thank you for the comments. You ask the author to add number of ‘female deaths per 100 person-years’ to make clear the difference. Even though it was stated in Table one of the manuscript, the authors incorporated it to the text part of the revised manuscript.

Reviewer concern: Also they should define what they intend with good, fair, and poor adherence to antiretroviral treatment.

Authors’ Response: 

• As stated above, the definition of ART drug adherence was incorporated to the method section of the revised manuscript. 

Reviewer concern: No data about baseline HIV-RNA load is present. I believe that this information is crucial both for deaths and virological failure. If this information is not available, the authors should discuss it in the limit's section.

Authors’ Response: 

• The reviewer also asked the author to include data about baseline HIV-RNA load and it is true that baseline HIV-RNA load is crucial both for deaths and virologic failure. 

• But, information related to viral load was not found on the majority of patients’ charts. 

• By this reason we were unable to incorporate it to our study; so, based on your suggestion we stated it as a limitation in the revised manuscript. 

Reviewer concern: The authors wrote, "From these, pre-ART duration, residence, functional status at baseline, and took Cotrimoxazole preventive therapy during follow up were found to be statistically significant predictors of the death rate among adults within the first year of ART initiation at a P-value of less than 0.05 (Table 3)". I think the authors mean "table 4". Check it?

Authors’ Response: 

• Actually it is table 3 not table 4 and the problem is on table 2 that I divided it to two by using the term “Table 2 (continued)”; but, in the revised manuscript table 2 is merged to a single table and the regression table remains as it is (i.e. table 3). 

Reviewer concern: 

• No data about the cause of death are present. I suggest adding them. 

• The presence of opportunistic infections at the moment of diagnosis is not present in the Cox model. Which was the CHR of having them?

• Also, CD4 cells/count does not been considered in the Cox model. 

• Different studies showed how late-presenters had increased mortality. I believe that the authors should discuss it.

Authors’ response: 

• The reviewer asked the authors to add data about the cause of death. But, it was already included in the previous manuscript entitled as “Predictors of mortality within one year of ART initiation”.

• As the reviewer stated, both the presence of opportunistic infections at the moment of diagnosis and CD4 cells/count are not presented in the Cox model. This is because of CHR of >0.2 in bi-variable analysis for both variables. As stated in the manuscript, the authors include variables to the Cox model only if the p-value at bi-variable analysis is less than 0.2. That is why these two variables are not included in the Cox model. 

• The reviewer asked the authors to discuss how late-presenters had increased mortality. But, it is already discussed in the previous manuscript with a variable name of “pre-ART duration (duration from HIV status confirmation to ART initiation)”. 

o It is also discussed in the document as: “Similarly, HIV infected patients who started ART six months and above after HIV status confirmation were 2.17 times (AHR=2.17; 95% CI: 1.24, 3.79) at higher risk of death within the first year than patients who started ART early.”

Table 1: 

Reviewer concern: Please recheck the percentages' approximation. The sum of "age in years", and Educational level" is 100.1% and 99.99% respectively.

Authors’ response:

• Thank you for your detail reviewing of this manuscript. The authors revised and corrected it into the correct percentage approximation and the sum became 100% for each variable.

Reviewer concern: 67% of people had a baseline CD4 cell count > 200 cells-mL and 41.9 more than 350. It is not clear why 65% of people took cotrimoxazole and 39.47% isoniazid. People who start isoniazid (how long did they take it?) had a positive tuberculin test?

Authors’ response: 

This is because some patients with confirmed HIV status are hesitated to start ART as early as they know their status. So in order to avoid the occurrence of OIs including TB, CPT was given for HIV patients whatever their CD4 count. Similarly, isoniazid also provided to them after Tb is ruled out.

Table 2: 

Reviewer concern: It is not clear what the authors mean with "Past OI", "Past CPT treatment", "Past INH prophylaxis", and "Past TB treatment history. Do they mean if patients have done the treatments before HIV diagnosis? The authors should include in the table a subtitle to explain all abbreviations used in the table.

Authors’ response: 

• Past OI = it is to mean presence of opportunistic infection/s after HIV diagnosis and before ART start.

• Past CPT treatment = it is to mean that "does a patient took cotrimoxazole preventive therapy (CPT) after HIV diagnosis and before ART start to prevent the occurrence of OIs"

• Past INH prophylaxis = means that "does a patient took Isoniazid (INH) prophylaxis after HIV diagnosis and before ART start to prevent the occurrence of tuberculosis?" for those who were ruled out from Tuberculosis disease. But, since the study was based on chart review, it was not informative whether TB was ruled out with tuberculin skin test or other tests; rather we simply took the record labeled as “TB status: positive or negative” irrespective of the type of laboratory performed. 

• Past TB treatment history = it is to mean that “did a patient took anti tuberculosis treatment before start of ART?”

• In the revised manuscript, a key is included to explain abbreviations in the tables.

Table 3.

Reviewer concern:

• Please remove "(continued)", or rename the table as table 2.

• About the initial ART regimens, the authors choose as variable "le" and "others". Please correct it. 

• Furthermore, I believe that it is interesting knowing what other regimens the patients started. In my opinion, this is important to understand why mortality is higher in "others" group.

• The authors should include in the table a subtitle to explain all abbreviations used in the table.

Authors’ response:

• The authors tried to merge the table to one table and removed the term “Continued”.

• In the revised manuscript 1e is edited as TDF-3TC-EFV and the reason that the authors categorize the initial ART regimens as "le" and "others" is that more than 94% of patients’ records showed that patients had took TDF-3TC-EFV and only 27 patients (5%) took other than TDF-3TC-EFV. Even, this 5% is constituted from the sum of five different combination therapies; namely: AZT + 3TC + EFV, TDF + 3TC + NVP, AZT + 3TC + LPV/r, AZT + 3TC + ATV/r, and TDF + 3TC + ATV/r. When these categories tabulated with the outcome variable, the assumption of chi-square was violated. So, to keep the assumption, the variable was re-categorized by merging these five combinations in to one (i.e. others). 

• In the revised manuscript, a key is included to explain abbreviations used in the table and drugs included in “others” category.

Table 4

Reviewer concern:

• It should be in the results section and not in the discussion

• The authors should include in the table a subtitle to explain all abbreviations used in the table.

• In table 2 the authors divided the educational level into "no education", "primary", "secondary", and college +". In this table, instead, they divided it into "no education", "10 education", and "20 & above". I suggest using the same division.

Authors’ response: 

• Thank you for the comments; actually it is table 3 not table 4 as I stated it before. 

• In the revised manuscript table 3 is sited in the results section, as per the comment. 

• Regarding educational level categorization: based on the comment I preferred to use common category in both tables (i.e. "no education", "10 education", and "20 & above") and corrected in the revised manuscript.

Figure 1: 

Reviewer concern: It is not clear what the number between brackets mean. Please comment.

Authors’ response: 

• The numbers in the brackets in Figure 1 is to show the number of deaths within the specified time interval. For example, number 20 in bracket is to mean that 20 deaths were occurred between time zero and 0.2 years. But, in order to avoid confusion the authors avoid these numbers in brackets at the revised manuscript. 

All figures: 

Reviewer concern:

• In the text, the authors wrote about mortality using "months of follow-up," while in the figures, time is expressed as a fraction of years. 

• It could create confusion in the readers because "0.6" could be interpreted as "6 months", instead of 7.2 months. 

• I suggest modifying the X-axes of the figures

• I suggest performing a Log-rank test to compare two samples' survival distributions in the different KM to see if there is a statistical difference.

Authors’ response:

• Thank you for your detailed viewing of the document. Actually, the x-axis of each figure was clearly entitled as “analysis time in year” so that “0.6” can be interpreted as 0.6 year. But to avoid a confusion the x-axis is labeled as “analysis time in month” in the revised manuscript, as per the comment. 

• Really, I accept the comment regarding performing a Log-rank test to see if there is a statistical difference between groups. Therefore, log-rank test value is analyzed and incorporated in the revised manuscript for each figure (fig2-5).

Thank you!!

---

## [Decision Letter · Decision Letter 1]

22 Feb 2021

PONE-D-20-34000R1

Incidence and predictors of mortality within the first year of antiretroviral therapy initiation at Debre-Markos referral hospital, Northwest Ethiopia: A retrospective follow up study

PLOS ONE

Dear Dr. Aemro,

Thank you for submitting your manuscript to PLOS ONE. After careful consideration, we feel that it has merit but does not fully meet PLOS ONE’s publication criteria as it currently stands. Therefore, we invite you to submit a revised version of the manuscript that addresses the points raised during the review process.

We look forward to receiving your revised manuscript.

Kind regards,

Giordano Madeddu

Academic Editor

PLOS ONE

Reviewers' comments:

Reviewer's Responses to Questions

**Comments to the Author**

1. If the authors have adequately addressed your comments raised in a previous round of review and you feel that this manuscript is now acceptable for publication, you may indicate that here to bypass the “Comments to the Author” section, enter your conflict of interest statement in the “Confidential to Editor” section, and submit your "Accept" recommendation.

Reviewer #1: All comments have been addressed

Reviewer #2: All comments have been addressed

2. Is the manuscript technically sound, and do the data support the conclusions?

Reviewer #1: Yes

Reviewer #2: Partly

3. Has the statistical analysis been performed appropriately and rigorously? 

Reviewer #1: Yes

Reviewer #2: Yes

4. Have the authors made all data underlying the findings in their manuscript fully available?

Reviewer #1: Yes

Reviewer #2: Yes

5. Is the manuscript presented in an intelligible fashion and written in standard English?

Reviewer #1: Yes

Reviewer #2: Yes

6. Review Comments to the Author

Reviewer #1: The paper is interesting and well written . Some data are missing but accordino to data sources missing.

Reviewer #2: I have re-read with interest the new version of the manuscript "Incidence and predictors of mortality within the first year of antiretroviral therapy initiation at Debre-Markos referral hospital, Northwest Ethiopia: A retrospective follow-up study". The authors provided to fix most of the issues present in the previous version. However, some issues still remain.

- Re-reading the manuscript I found many typos (e.g. Efavirinz).

- Abbreviation should be written entirely in the first appearance in the text, even if used in the abstract.

- In the previous revision, I ask the authors to discuss the relationship between late presents and mortality in their cohort. The authors reply, "it is already discussed in the previous manuscript with a variable name of “pre-ART duration (duration from HIV status confirmation to ART initiation)”. However, the duration between HIV confirmation and ART initiation does not concern the "late-presenters status", that it is defined as a CD4 cell count <350cells/mm3, without having opportunistic infections. I suggest the authors read this paper 10.1111/j.1468-1293.2010.00857.x, written by the European Late Presenter Consensus working group.

- In my previous revision, I suggested adding more information about the cause of death in the cohort. The authors replied that ”it was already included in the previous manuscript entitled as “Predictors of mortality within one year of ART initiation”.” In my opinion, these results must also be reported in this manuscript because they are crucial for the reader, and he/she does not have to search and read another paper to understand the cause of death.

- The authors reply satisfactorily about the meaning of past opportunistic infection, past CPT treatment, past INH prophylaxis, past TB treatment history. However, I suggest giving this information not only to me, but also to the reader, adding these explanations in the methods section.

- The authors reply that the presence of opportunistic infections at the moment of diagnosis and CD4 cells/count are not presented in the Cox model. This is because of CHR of >0.2 in bi-variable analysis for both variables. In my opinion, these results are important and need to be discussed because AIDS-presenters normally have an increased risk of death. Furthermore, I suggest adding all this statistical analysis as supplemental material.

- The authors wrote that many of the patients had a hesitation on starting the antiretroviral treatments. Could it be due to a lack of communication between the medical staff and the patients? Do you think that longer counseling could reduce this hesitation? Do you normally talk with the patients about U=U (undetectable = untransmittable)? Calabrese et al., in their paper, published in “the lancet HIV, explain why providers should discuss U=U with the patients https://doi.org/10.1016/S2352-3018(19)30030-X. Furthermore, a recent analysis on real-life patients reinforced the validity of this message https://doi.org/10.1097/QAD.0000000000002825. In my opinion, discussing this aspect in the Ethiopian context would increase the value and the originality of the work.

7. PLOS authors have the option to publish the peer review history of their article (what does this mean?). If published, this will include your full peer review and any attached files.

Reviewer #1: No

Reviewer #2: No

---

## [Author Response · Author response to Decision Letter 1]

21 Apr 2021

March 20, 2021

PLOS ONE editors

Journal of PLOS ONE

Dear PLOS ONE editors

Subject: Submission of revised manuscript entitled as “Incidence and predictors of mortality within the first year of antiretroviral therapy initiation at Debre-Markos referral hospital, Northwest Ethiopia: A retrospective follow up study”, after second revision.

EMID: 57e8cf222f91fd0f

Thank you for email dated on Feb 22, 2021, enclosing the reviewer’s comments. We have carefully revised the manuscript and incorporated their comments accordingly. Our responses are given in point-by-point response below.

We hope the newly revised version is suitable for publication and look forward to hearing from you in due courses.

Sincerely!

Agazhe Aemro Terefe (Corresponding author)

University of Gondar, College of Medicine and health Sciences, School of Nursing, Department of Medical Nursing.

Point by point response to reviewers’ comments

Part I: Editors’ concerns and response by the authors:

Dear editors, the authors would like to thank you for your comment and your concern to this manuscript. 

The authors tried to ensure that the manuscript meets PLOS ONE's style requirements.

The newly revised manuscript contains:

o A rebuttal letter that responds to each point raised by the academic editor and reviewer(s).

o A marked-up copy of the manuscript that highlights changes made to the original version.

o The new version of the revised manuscript without tracked changes. 

Since it is not applicable, the authors didn’t incorporate the laboratory protocol. 

Part II: Reviewers’ concerns and response by the authors:

Reviewer #1: 

Thank you for your in-depth and detailed viewing of the manuscript entitled as “Incidence and predictors of mortality within the first year of antiretroviral therapy initiation at Debre-Markos referral hospital, Northwest Ethiopia: A retrospective follow up study”. Your comments inspires the authors to do more in order to have an interesting paper. We hope this manuscript will be considered until publication.

Reviewer #2: 

Thank you for your full interest in reading the revised manuscript and for your constructive comments regarding the manuscript. Based on your comments, the authors will tried to fix the remaining issues.

Reviewer concern: 

Re-reading the manuscript I found many typos (e.g. Efavirinz).

Abbreviation should be written entirely in the first appearance in the text, even if used in the abstract.

Authors’ response: 

Yes, the Authors accept that there are typing errors in the manuscript and we tried to correct and re-type the error in the newly revised manuscript.

As per the comment, the authors wrote abbreviations entirely in the first appearance. 

Reviewer concern: 

In the previous revision, I ask the authors to discuss the relationship between late presents and mortality in their cohort. The authors reply, "it is already discussed in the previous manuscript with a variable name of “pre-ART duration (duration from HIV status confirmation to ART initiation)”. However, the duration between HIV confirmation and ART initiation does not concern the "late-presenters status", that it is defined as a CD4 cell count <350cells/mm3, without having opportunistic infections. I suggest the authors read this paper 10.1111/j.1468-1293.2010.00857.x, written by the European Late Presenter Consensus working group. 

Authors’ response:

Sorry for the misunderstanding; in the newly revised manuscript, the authors tried to show the relation between late presentation and mortality among HIV-infected patients. Also, it is clearly stated in the introduction part from line 92 to 98 (page 5).

Reviewer concern: 

In my previous revision, I suggested adding more information about the cause of death in the cohort. The authors replied that “it was already included in the previous manuscript entitled as “Predictors of mortality within one year of ART initiation”.” In my opinion, these results must also be reported in this manuscript because they are crucial for the reader, and he/she does not have to search and read another paper to understand the cause of death.

Authors’ response:

Based on the comment, the authors tried to review different kinds of literatures and identified predictors of early mortality stated by different literatures. The authors incorporated these details in the newly revised manuscript (see line 99 to 105 on page 5).

Reviewer concern: 

The authors reply satisfactorily about the meaning of past opportunistic infection, past CPT treatment, past INH prophylaxis, past TB treatment history. However, I suggest giving this information not only to me, but also to the reader, adding these explanations in the methods section.

Authors’ response:

Based on the comment, the authors incorporated these definitions to the newly revised manuscript at the method section, particularly at the operational definition (see from line 149 to 156 on page 7-8).

Reviewer concern: 

The authors reply that the presence of opportunistic infections at the moment of diagnosis and CD4 cells/count are not presented in the Cox model. This is because of CHR of >0.2 in bi-variable analysis for both variables. In my opinion, these results are important and need to be discussed because AIDS-presenters normally have an increased risk of death. Furthermore, I suggest adding all this statistical analysis as supplemental material. 

Authors’ response:

Based on the comment, the authors discussed this issue clearly in the newly revised manuscript (see from line 283 to 285 on page 15). 

Furthermore, the authors uploaded all the bi-variable analysis table as supplemental material.

Reviewer concern: 

The authors wrote that many of the patients had a hesitation on starting the antiretroviral treatments. Could it be due to a lack of communication between the medical staff and the patients? Do you think that longer counseling could reduce this hesitation? Do you normally talk with the patients about U=U (undetectable = untransmittable)? Calabrese et al., in their paper, published in “the lancet HIV, explain why providers should discuss U=U with the patients https://doi.org/10.1016/S2352-3018(19)30030-X. Furthermore, a recent analysis on real-life patients reinforced the validity of this message https://doi.org/10.1097/QAD.0000000000002825. In my opinion, discussing this aspect in the Ethiopian context would increase the value and the originality of the work.

Authors’ response:

Thank you for the link you gave to the authors to read more regarding the issue stated in this part. 

Based on the comment you gave, the authors explained the reason why HIV-infected patients hesitate to start ART as early as possible in Ethiopian context.

In the revised manuscript, the authors incorporated the implication of counseling and educating patients about U=U (undetectable = un-transmittable). (See from line 59 to 67 on page 3-4).

Thanks too much!

---

## [Editor Report · Decision Letter 2]

30 Apr 2021

Incidence and predictors of mortality within the first year of antiretroviral therapy initiation at Debre-Markos referral hospital, Northwest Ethiopia: A retrospective follow up study

PONE-D-20-34000R2

Dear Dr. Aemro,

We’re pleased to inform you that your manuscript has been judged scientifically suitable for publication and will be formally accepted for publication once it meets all outstanding technical requirements.

Kind regards,

Giordano Madeddu

Academic Editor

PLOS ONE
---

## [Editor Report · Acceptance letter]

6 May 2021

PONE-D-20-34000R2 

Incidence and predictors of mortality within the first year of antiretroviral therapy initiation at Debre-Markos referral hospital, Northwest Ethiopia: A retrospective follow up study 

Dear Dr. Aemro:

I'm pleased to inform you that your manuscript has been deemed suitable for publication in PLOS ONE. Congratulations! Your manuscript is now with our production department. 

Kind regards, 

on behalf of

Dr. Giordano Madeddu 

Academic Editor

PLOS ONE